# HERC5 and the ISGylation Pathway: Critical Modulators of the Antiviral Immune Response

**DOI:** 10.3390/v13061102

**Published:** 2021-06-09

**Authors:** Nicholas A. Mathieu, Ermela Paparisto, Stephen D. Barr, Donald E. Spratt

**Affiliations:** 1Gustaf H. Carlson School of Chemistry and Biochemistry, Clark University, 950 Main St., Worcester, MA 01610, USA; nmathieu@clarku.edu; 2Department of Microbiology and Immunology, Schulich School of Medicine and Dentistry, Western University, 1151 Richmond St., London, ON N6A 5C1, Canada; epaparis@uwo.ca (E.P.); sbarr9@uwo.ca (S.D.B.)

**Keywords:** ISG15, ISGylation, HECT E3 ubiquitin ligase, HERC5, antiviral immune response, interferon

## Abstract

Mammalian cells have developed an elaborate network of immunoproteins that serve to identify and combat viral pathogens. Interferon-stimulated gene 15 (ISG15) is a 15.2 kDa tandem ubiquitin-like protein (UBL) that is used by specific E1–E2–E3 ubiquitin cascade enzymes to interfere with the activity of viral proteins. Recent biochemical studies have demonstrated how the E3 ligase HECT and RCC1-containing protein 5 (HERC5) regulates ISG15 signaling in response to hepatitis C (HCV), influenza-A (IAV), human immunodeficiency virus (HIV), SARS-CoV-2 and other viral infections. Taken together, the potent antiviral activity displayed by HERC5 and ISG15 make them promising drug targets for the development of novel antiviral therapeutics that can augment the host antiviral response. In this review, we examine the emerging role of ISG15 in antiviral immunity with a particular focus on how HERC5 orchestrates the specific and timely ISGylation of viral proteins in response to infection.

## 1. Introduction

Viral replication is a highly coordinated process that relies heavily on the host cellular machinery and can disrupt critical cellular activities leading to disease and/or death [1,2,3,4]. Viruses exert immense evolutionary pressure on their hosts, driving the development of complex cellular immune responses that work to identify, combat and eliminate infectious pathogens. For example, various mammalian tissues contain immune cells that use pattern recognition receptors (PPRs), including toll-like receptors (TLRs), NOD-like receptors (NLRs) and RIG-1-like receptors (RLRs), to screen for pathogen-associated molecular patterns (PAMPs) in the cytosol and extracellular matrix [5,6]. When cells detect a viral pathogen, specific PRRs become activated that signal for mitochondrial antiviral signaling (MAVS) proteins to upregulate interferon-α/β (IFN-α/β) cytokine production [2,7,8,9]. Cells then secrete IFN-α/β cytokines into the extracellular matrix to alert naive cells of an impending infection [10,11].

The production of IFN-α/β is one of the first lines of cellular defense against viral pathogens. IFN-α/β activates the Janus kinase-signal transducer and activation of transcription (JAK–STAT) signaling pathway (Figure 1). When IFN-α/β binds to an IFN-specific receptor on the cellular surface, a conformational change occurs that exposes sites on the cytoplasmic part of the receptor that become phosphorylated by Janus kinase 1 (JAK1) and tyrosine-protein kinase 2 (TYK2). This phosphorylation attracts the nuclear transcriptional regulators signal transducer and activator of transcription proteins 1 and 2 (STAT1 and STAT2) to the cellular membrane where they are also phosphorylated by TYK2. The phosphorylated STAT1 and STAT2 proteins form a ternary complex with methylated interferon regulatory factor 9 (IRF9), which is subsequently demethylated to signal for the migration of the STAT1–STAT2–IRF9 (SSI) complex to the nucleus. In the nucleus, the SSI complex binds to the ISG promoter region of the interferon-stimulated response element (ISRE), resulting in the transcriptional upregulation of hundreds of ISGs that augment the antiviral immune response by halting ribosomal protein synthesis [12,13], inducing the activation of regulatory cell death pathways [14,15,16,17,18], and carrying out the post-translational modification (PTM) of host and viral proteins that impact viral replication [19,20,21,22,23,24,25].

HECT and RCC1-containing protein 5 (HERC5) and interferon-stimulated gene 15 (ISG15) are two antiviral immune proteins that are induced following IFN-α/β signal transduction. Multiple studies have established that HERC5 plays a central role in mammalian innate immunity by ISGylating viral proteins to disrupt viral replication [26,27]. The antiviral function of HERC5 was first discovered by researchers who found that its co-expression with the E1 ubiquitin-like modifier activating enzyme UBE1L (Uba7; E.C. 6.2.1.45) and UBE2L6 (UbcH8; E.C. 2.3.2.23) resulted in normal rates of cellular ISGylation activity, despite the absence of IFN-α/β stimulation [28].

The functional characterization of phosphokinase signal transduction pathways has now clarified that UBE1L, UBE2L6, HERC5 and ISG15 expression are predominantly induced by the JAK–STAT signaling pathway (Figure 1) [29,30]. However, in recent years, additional modes of ISG induction have been identified in cells. For example, the ISG15-specific cascade enzymes are thought to be induced by interferon regulatory factors 3, 5 and 7 (IRF3, IRF5 and IRF7) [31,32]. HERC5 expression is also regulated by the cytokine interleukin 1 β (IL-1β) and tumor necrosis factor α (TNFα) in distinct signal transduction pathways. Moreover, recent studies have shown that ISG15, UBE1L, UBE2L6 and HERC5 are tightly regulated by nuclear factor kappa-light-chain-enhancer of activated B cells (NF-κB)–an immunological protein complex that is involved in controlling diverse cellular responses to viral infection including cytokine production, DNA transcriptional regulation and interferon activation [33].

## 2. The ISGylation Pathway

ISGylation involves the covalent posttranslational attachment of ISG15 to host and viral target substrates by a specialized group of IFN-α/β induced E1-E2-E3 ubiquitin cascade enzymes. For example, protein–protein interaction studies have shown that ISG15 only coordinates with five of the over 600 identified E1–E2–E3 enzymatic members [34]. The ISGylation pathway begins when UBE1L activates ISG15 through an ATP-dependent mechanism to form a thioester bond between the *C*-terminal carboxyl of ISG15 and the catalytic cysteine of UBE1L [35,36] (Figure 2). Following activation, ISG15 is transferred by UBE1L to UBE2L6 via a trans-thiolation reaction that forms a thioester bond between the *C*-terminus of ISG15 and the conserved catalytic cysteine residue of UBE2L6 [37,38]. The UBE2L6–ISG15 thioester complex then transfers its ISG15 cargo to the catalytic cysteine (C994) of the ISG15-specific homologous to E6AP *C*-terminus (HECT) E3 ligase HERC5 (EC 2.3.2.-). Thereafter, HERC5 catalyzes the covalent attachment of ISG15 onto host and viral substrate proteins by forming an isopeptide bond between the *C*-terminus of ISG15 and the ε-amino group on the lysine of the respective protein target [27,28].

Studies have shown that the really interesting new gene (RING) E3 ligase tripartite motif-containing protein 25 (TRIM25; EC: 2.3.2.27) and the RING-between-RING (RBR) E3 ubiquitin ligase human homolog of Ariadne (HHARI) can also mediate the conjugation of ISG15 to cellular substrates [39,40]. TRIM25 achieves this catalytic activity by using its *N*-terminal RING domain as a scaffold to coordinate the transfer of ISG15 from UBE2L6 to the target protein via the ubiquitylation mechanism used by other members of the RING E3 ubiquitin ligase family [39]. Likewise, HHARI catalyzes the post-translational attachment of ISG15 to its target substrates using a canonical RBR ubiquitylation mechanism [40]. In this model, HHARI catalyzes substrate ISGylation by using a RING-like mechanism to coordinate ISG15-charged UBE2L6, followed by the formation of a HECT-like thioester intermediate between ISG15 and the HHARI required for the catalysis domain (Rcat; RING2) to complete the ISG15 transfer onto the target substrate [40].

Studies on the ubiquitylation pathway have found that E1–E2–E3 cascade enzymes are constitutively expressed and can form a variety of ubiquitin chain linkages with their target substrates [41,42,43,44,45]. Interestingly, the ISG15-specific E1–E2–E3 cascade enzymes are only induced following IFN-α/β signal transduction and exclusively form monomer isopeptide linkages between their ISG15 cargo and target substrate [46]. Accordingly, the ISG15 signaling pathway provides the cell with a highly specialized antiviral function at the expense of regulating vast cellular activities [47]. Intriguingly, a recent publication observed that hybrid ISG15–ubiquitin chains can also be formed by the cell to regulate protein homeostasis [48]. As novel ISG15 signaling mechanisms continue to be discovered, it will be important to refine the current understanding of how UBE1L, UBE2L6 and HERC5 target host and viral substrates for protein ISGylation. 

## 3. ISG15: A Critical Moderator of the Host Antiviral Response

The rapid induction of ISG15 in response to viral infection has significant impacts on the cellular environment. As such, tight regulation of the ISGylation signaling pathway is necessary for cell survival. For example, reduced expression of the ISG15-dependent ubiquitin-specific protease 18 (USP18) results in the hyper-ISGylation of cellular proteins, induction of apoptosis in hemopoietic tissues, brain cell injury and decreased life expectancy in murine models [49]. ISG15 deficiencies in humans are extremely rare and tend not to be fatal; however, such deficiencies have been associated with mycobacterial hypersensitivity, brain calcification and skin lesions [49,50,51,52]. Therefore, understanding how ISG15 influences cellular function during a viral infection is essential to clarifying the diverse range of biochemical outcomes that can occur as a consequence of host ISGylation activity.

To date, most studies have shown that cellular ISGylation activity, albeit inhibitory on the function of other proteins, does not signal for host or viral protein degradation as would a K48 poly-ubiquitin chain [25,53]. Rather, the attachment of ISG15 to host and viral substrates results in a range of stabilizing effects that are similar to the function of K63 di-ubiquitin chain signaling [54,55,56,57,58,59]. The cellular effects caused by protein ISGylation depends on the stage of viral infection and what specific substrate proteins are being targeted by the ISG15 cascade. For example, cellular ISGylation activity has been shown to have an antiviral effect by (i) directing the cytosolic localization of viral proteins to inhibit their reassembly and production and (ii) stabilizing key antiviral host proteins to augment the collective immune response [60]. In contrast, studies have also reported that the ISGylation of host proteins can reduce the potency of the cellular immune response during the later stages of viral infection by signaling for host protein inhibition and/or degradation [46].

A 2003 study conducted by Malakohova et al. was the first to identify that ISG15 positively regulates the host antiviral response. Their work determined that 1-phosphatidylinositol 4,5-bisphosphate phosphodiesterase gamma 1 (PLCγ1), mitogen-activated protein kinase 3 (MAPK3, aka ERK-1), JAK-1, and STAT1 are cellular ISGylation targets [61]. Specifically, the ISGylation of these proteins was shown to upregulate the expression of ISG15-conjugating enzymes by stabilizing proteins involved in the IRF9–JAK–STAT transduction pathway [61]. These findings were later supported by Przanowski et al. who showed that (i) ISGylation activity in lipopolysaccharide (LPS)-stimulated microglia promoted STAT1 stabilization in the IRF9 signal transduction pathway and (ii) cellular ISG protein levels increased following the ISGylation of STAT1 [62].

A recent study by Shi et al. has found that the ISG signal transduction protein IRF3 is targeted for ISGylation by the E3 ligase HERC5 [63]. Analogous to IRF9, IRF3 is a master regulatory protein that activates ISG gene transcription following the intracellular detection of viral PAMPs by PRRs, or through the host recognition of viral RNA by the cyclic GMP–AMP synthase/stimulator of interferon genes protein (cGAS/STING) pathway [64,65]. Under this mode of ISG induction, IRF3 undergoes a phosphorylation-dependent dimerization event that triggers its translocation to the nucleus. Upon nuclear entry, the IRF3 complex acts as a transcription factor of type I interferons and other proinflammatory genes related to the ISG family. The antiviral potency of IRF3 requires a finely tuned response to environmental signals, as is evident by the multitude of post-translational modifications that are required to control IRF3 activation and downstream cellular function. For example, HERC5 and ISG15 work together to modulate IRF3 activity by catalyzing the ISGylation of IRF3 at K193, K360 and K366 residues [63]. Once ISGylated by HERC5, IRF3 is no longer targeted by the RING E3 ligase peptidyl-prolyl cis-trans isomerase 1 (Pin1) for K48 polyubiquitylation. Consequently, ISGylated IRF3 is able to circumvent 26S-proteosomal degradation, which increases ISG induction rates in virally infected cells [63]. Taken together, these studies suggest that HERC5 and ISG15 act as positive regulators of the ISGylation feedback loop by (i) amplifying the induction of ISGs during the early stages of viral infection, and (ii) preventing the premature termination of the antiviral interferon response by enhancing IRF-related signal transduction.

New studies have discovered additional viral and host target substrates for the ISG15-specific enzymatic cascade. For example, in 2019 Zhang et al. used quantitative labile free proteomics to identify new ISGylation targets in mouse liver cells infected by the bacteria *Listeria monocytogenes* [66]. Proteomic analysis confirmed 87 previously identified ISGylation targets and 347 new substrate proteins that are targeted by ISG15-specific cascade enzymes [66]. For example, UBE2L6 and ISG15 were determined to ISGylate retinoic acid-inducible gene I (RIG-I) and the RNA helicase melanoma differentiation-associated protein 5 (MDA5)–PRRs that signal for the activation of immunoproteins involved with the humoral antiviral host response [67]. The functional result of RIG-I ISGylation was discovered by Arimoto et al., who determined that ISGylated RIG-I is ubiquitylated by RING finger protein 125 (RNF125) to block RIG-I cellular signaling [67]. Intriguingly, studies by Kim et al. also propose that ISGylated RIG-I and MDA5 do not undergo TRIM25 mediated K63-polyubiquitylation as they would under normal cellular conditions, but rather undergo K48 polyubiquitylation by RNF125 and are subsequently targeted for 26S proteasomal degradation [68]. Collectively, these findings verify that the ISGylation of RIG-I and MDA5 suppresses the antiviral host response by reducing cellular interferon production through an ubiquitylation-dependent mechanism [69].

ISG15 and its specific cascade enzymes are also thought to inhibit the host antiviral immune response under certain cellular conditions. For example, new studies propose that high cellular ISG levels can interrupt the NF-kB pathway and hinder the host proinflammatory response [46]. It has also been found that the phosphorylation of nuclear receptor subfamily 2 group C member 2 (also known as TAK1) leads to the ISGylation, ubiquitylation and inactivation of UBE2N (Ubc13), an important E2 ubiquitin-conjugating enzyme involved in promoting the activity of immunologically aggressive pathways related to host protection [46]. Similarly, the ISGylation pathway negatively regulates the autophagic activity of macrophages via the ISGylation of beclin-1 (BECN1) following type I interferon induction [70,71].

Mounting experimental evidence has demonstrated that ISG15 and its associated ISG15-specific cascade enzymes dictate the expression and activity of immunological host proteins during the antiviral response. In biologically relevant systems, the induction of the ISG15-specific enzymatic cascade has been shown to regulate both host and pathogenic proteins. Together, these ISGylation processes are dependent on the concentration of viral proteins and RNA present in the cell, the expression levels of ISGs and the temporal progression of the immune response as part of the interplay between the host and viral machinery. During the early stages of infection, it appears that the ISGylation of key immune factors serves to increase the production of antiviral protein factors to ensure the timely activation of the host’s innate interferon system. Later, as the host’s immune response begins to control the viral infection, it is possible that the ISGylation pathway may lead to a reduction in the expression, stability and activity of host immunoproteins to prevent the risk of chronic inflammation, autoimmunity and ensuing tissue damage.

## 4. HERC5: A Unique HECT E3 Ubiquitin Ligase That ISGylates Viral Proteins

HERC5 is a multidomain 114 kDa protein that belongs to both the HECT E3 ubiquitin ligase family and the RCC1 superfamily [72]. To date, 28 HECT E3 ubiquitin ligases have been identified in humans, with each having a conserved HECT domain found near their *C*-termini [73,74,75]. The HECT domain consists of two lobes—the *N*-terminal lobe that contains the binding site for E2 cognate enzymes, and the *C*-terminal lobe that contains the E3′s absolutely conserved catalytic cysteine [76]. Coordination of the HERC5 *C*-terminal lobe to ISG15, as opposed to ubiquitin, provides HERC5 with a potent anti-viral activity that is achieved by the covalent attachment of ISG15 to viral proteins. Structural studies on members of the HECT family have also revealed that there is a flexible hinge region located between the *N*- and *C*-terminal lobes of the HECT domain [77,78,79,80,81]. In the context of ISGylation, this inherent flexibility appears to be critical for allowing the juxtaposition of the catalytic cysteines in UBE2L6 and HERC5 for efficient ISG15 transfer [82], as well as for moving the ISG15-charged HECT *C*-terminal lobe of HERC5 into the proximity of viral substrates [83]. Previous domain-mapping experiments have determined that the HERC5 *N*-terminal region contains five regulator of chromatin condensation 1 (RCC1) motifs which collectively form an RCC1-like domain (RLD)—a substructure of HERC5 that plays a direct role in the recognition and coordination of viral substrates [84]. For example, studies have shown that HERC5 uses its RLD domain to bind the influenza A virus (IAV) non-structural protein 1 (NS1) [47], the hepatitis C virus (HCV) non-structural protein 5A (NS5A) [85] and multiple human immunodeficiency virus (HIV) gag particle precursor proteins [86]. Given the expansive amount of new research being conducted on HERC5 ISGylation, it is likely that additional HERC5 pathogenic substrates will be discovered, and the immunological consequences of these interactions will need to be resolved.

HERC5 is the only human HECT E3 ubiquitin ligase that is known to conjugate ISG15 to host and viral substrates. The immunological significance of HERC5 has become well established, with recent studies beginning to reveal what proteins are targeted by the enzyme during HCV, IAV, HIV-1 and other viral replication cycles, and whether viral protein activities are inhibited or augmented following ISG15 attachment (Table 1).

Sequence alignment studies have found that the HERC5 domain architecture is paralogous to another HECT E3 ubiquitin ligase, HERC6, but this enzyme demonstrates a limited capability to coordinate ISG15 substrate transfer except in murine models [108]. These findings are surprising, particularly with the high level of sequence conservation between HERC5 and HERC6, and thus leaving many questions on the structural mechanisms used by HERC5 to mediate viral protein ISGylation. To uncover these mysteries, new research is required to clarify how HERC5 use its *N*-terminal RLD domain to recognize and bind a wide range of viral substrates. Furthermore, now that there is a consensus as to what structures and mechanisms are used by HERC5 to attach ISG15 onto viral substrates, it will be important to explore the molecular dynamics involved with HERC5-substrate targeting, as well as how these dynamics allow for certain viral protein antagonists to subvert ISG-related immune responses.

HERC5 has recently been shown to carry out viral protein ISGylation to (i) stall IAV replication by blocking NS1 protein homodimerization [47], (ii) promote HCV proliferation via improved cyclophilin A recruitment by NS5A proteins [85] and, alternatively, (iii) inhibit an early stage of HIV assembly by attenuating gag-particle production [86,109]. In the upcoming sections we discuss some of the recent discoveries involving the HERC5-dependent ISGylation of viral protein targets.

### 4.1. HERC5-Dependent ISGylation of HCV

The antiviral effects of HERC5 and ISG15 on HCV were first observed by Jung Kim et al., who were examining mechanisms of HCV replication cycle inhibition [110]. By conducting site-directed mutagenesis experiments on the HCV protein NS5A, the group was able to demonstrate that HERC5 inhibits HCV by ISGylating NS5A at lysine 379 (K379) to result in NS5A becoming targeted for K48 polyubiquitylation by an unknown E3 ligase [110]. The researchers confirmed this antiviral activity by demonstrating that HCV replication was unrestrained when ISG15 and its corresponding E1–E2–E3 enzyme cascade were expressed in the presence of HCV NS5A K379R variant proteins. Taken together, these findings suggest that NS5A is the primary HCV target substrate of HERC5-dependent ISGylation, and that K379 is the sole NS5A residue that HERC5 targets for ISGylation. More recently, Abe et al. demonstrated that the HCV NS5A protein is prone to ISGylation at five Lys residues (K44, K68, K166, K215 and K308) [85]. In fact, one of the ISG15 attachment points on NS5A, K308, is located within the NS5A-binding region for cyclophilin A (CypA), a virulence factor that is required for efficient HCV cellular propagation. Thus, HERC5 ISGylation of NS5A at K308 enhances HCV recruitment of CypA to provide a pro-virulent replication activity, a direct contradiction of previous findings from Kim et al. [110]. Moreover, a stand-alone study conducted by Domingues et al. found that ISG15-related forms of HCV inhibition occur independently of HERC5 activity [111]. To date, no follow up research has been conducted on the HERC5–HCV ISGylation system to confirm whether HERC5 ISGylation activity results in a pro- or anti-virulent response to HCV infection, and whether HERC5 is necessary for catalyzing the modes of ISG15 inhibition that have been observed for HCV NS5A proteins.

### 4.2. Influenza Viruses and ISGylation by HERC5

New research by Tang et al. has shown that HERC5 targets IAV NS1 for ISGylation to prevent IAV capsid formation [47]. Using pulldown assays and immunoprecipitation analysis, the researchers found that HERC5 interacts with the ribosomal-binding (RBD) and C-terminal effector domains (ED) of NS1, and that both interactions were required to form stable HERC5–NS1 complexes [47]. Lysine residue substitutions in IAV NS1 also revealed that HERC5 attaches ISG15 at multiple NS1 lysine residues (K20, K41, K217, K219, K108, K110 and K126), with the strongest inhibitory effect coming from the ISGylation of the K126 and K217 residues in the ED and RBD domains, respectively [56]. The ISGylation of NS1 subsequently abolished the ability of NS1 to interact with protein kinase R (PKR) and blocked NS1 RBD-dependent homodimerization, thereby significantly inhibiting IAV capsid assembly in vivo [47]. Interestingly, it was found that certain avian flu IAV strains, such as H5N1, demonstrated a higher rate of NS1 K217R mutation compared to most seasonal flu strains [47,112]. It was also observed that avian IAV variants were less susceptible to ISGylation at K126, suggesting that, unlike other IAV strains, avian IAV NS1 proteins may adopt a new structure that obstructs the K126 ISGylation site from HERC5. Given that K126 and K217 have been determined as the primary ISGylation sites involved with preventing IAV capsid formation, these discoveries provide a possible explanation for why avian strains of the flu are more infectious and lethal than other IAV strains. Despite HERC5 demonstrating minimal antiviral activity against certain avian IAV strains, these findings from Tang et al. indicate that HERC5 and its ISGylation activity could serve as prime drug targets for the development of treatments that are used to combat most seasonal IAV strains.

### 4.3. HERC5 Modes of Action Against HIV

A pivotal study by Woods et al. in 2011 confirmed that HERC5 prevents an early stage of HIV-1 viral assembly by ISGylating proteins involved with Gag polyprotein (Pr55Gag) particle production [86]. The researchers used confocal immunofluorescence microscopy to reveal that IFN-I-induced HERC5 localizes to the cytoplasm where it forms punctuate bodies in a variety of cell lineages, and that these bodies interact with polyribosomes [86]. These findings were consistent with a previous study conducted by Durfee et al., who observed that HERC5 associates with the 60S ribosomal subunit of the polyribosome using cell fractionation [113]. Additionally, Woods et al. found that IFN-I-induced HERC5 and HIV Gag proteins colocalize to the plasma membrane where HERC5 ISGylates Gag particles to prevent HIV-1 viral budding. Expanding on these findings, the research group showed HERC5 possesses a second distinct mode of HIV inhibition that is independent of its HECT ISGylation activity [60]. Typically, in eukaryotic cells, nascent RNA molecules remain in the nucleus until their introns are spliced to reach maturation. However, HIV-1 can circumvent this cellular activity by expressing the nuclear trans-activator protein Rev, which binds to the Rev-response element (RRE) located within the HIV-1 intron [114]. These Rev–RRE nuclear interactions permit the nuclear export of unspliced HIV-1 RNA constructs via a translocation mechanism that involves exportin-1 (CRM1) and Ras-related nuclear protein guanidine triphosphate (RanGTP) [114]. Focusing specifically on mechanisms involved with HIV transcription, Woods et al. determined that the HERC5 RLD domain localizes to the perinuclear area and reduces intracellular RanGTP levels [60]. This correlated with the subcellular mis-localization of Rev/RRE following their nuclear export. These findings led Woods et al. to conclude that HERC5 acts as a host restriction factor during HIV replication by preventing an early stage of HIV-1 Pr55Gag particle production through a HECT-dependent mechanism, and that HERC5 affects nuclear export of unspliced HIV RNA constructs via an RLD-dependent mechanism [60]. Importantly, the same research group has shown that other retroviruses such as murine leukemia virus (MLV) and simian immunodeficiency virus (SIV) are also inhibited by HERC5 [86]. Interestingly, although SIV was inhibited by human HERC5, an ancestral version of HERC5 found in coelacanth fish was unable to inhibit SIV replication, suggesting that the HERC5 gene has evolved to combat lentiviruses in primates [115].

### 4.4. HERC5 Combating Other Viruses

New reports have suggested that HERC5 and ISG15 play a significant role in preventing the replication of many viral pathogens; however, the modes of viral inhibition exhibited by HERC5 and ISG15 on these pathogens are still being characterized. For example, HERC5 has been shown to ISGylate the major capsid protein L1 of human papilloma virus 16 (HPV16) to reduce the rate of viral budding, but the specific protein interactions that occur between HERC5, ISG15 and the L1 capsid protein of HPV16 are unknown [113]. Another study conducted by Kim et al. found a similar function for HERC5 and ISG15 during cytomegalovirus (CMV) replication, whereby viral budding was halted at the cellular membrane via the HERC5-dependent ISGylation of multiple CMV factors [93]. It was also observed that HERC5 and ISG15 are upregulated in response to infection caused by the Zika virus [116,117,118,119], Ebola virus [120,121,122,123], vaccinia virus [107], Kaposi’s sarcoma-associated herpesvirus (KSHV) [101], influenza B virus [124] and others (please see Table 1); however, the specific structural and biochemical modes of inhibition enacted by HERC5 and ISG15 in response to these pathogens is not fully understood.

Interestingly, new studies on the Ebola virus have shown that HERC5 and ISG15 exert two independent mechanisms of inhibition. These studies show that HERC5 reduces intracellular Ebola virus RNA through an RLD-dependent mechanism or, alternatively, free ISG15 can inhibit viral budding at the plasma membrane [125]. ISG15 specifically inhibits VP40 budding at the plasma membrane by blocking neural precursor cell-expressed developmentally down-regulated protein 4 (NEDD4)-mediated ubiquitylation of the VP40 late domain—a PTM that usually signals for the host endosomal sorting complexes required for transport (ESCRT) machinery to initiate budding of viral particles [125].

Studies in mice have found that vaccinia virus replication is enhanced in ISG15 knockout cells and inhibited in cells expressing wildtype levels of ISG15 [107]. ISG15-deficient cells were more resistant to apoptosis and had impaired phagocytic activity when coming in contact with infected cells [105]. These studies demonstrated that control of vaccinia virus is dependent on the level of cellular ISGylation activity, where either ISG15 knockout or the deISGylation of proteins can lead to enhanced infection kinetics.

Although it is well established that ISG15 and the ISGylation cascade enzymes are induced by cytomegalovirus infection, their role and direct interaction with viral proteins were only recently described [93]. Kim et al. demonstrated that knocking down ISG15 or HERC5 lead to a significant increase in viral titers, whereas expression of ISG15 and HERC5 inhibited viral replication by reducing viral gene expression and viral budding at the plasma membrane [93]. To date, much of the research into cytomegalovirus infection and its inhibition by HERC5 and ISG15 has focused on human cytomegalovirus (HCMV) protein antagonism. However, little is known about the specific effect that ISGylation has on viral proteins other than it is has a dominant inhibitory effect. Further studies are needed to elucidate the precise mechanism of HCMV inhibition demonstrated by HERC5 and ISG15.

HERC5 has been shown to inhibit KSHV in an ISGylation-dependent manner, with the knockdown of either HERC5 or ISG15 prior to viral reactivation resulting in higher titers of infectious viral particles [101]. For example, ISG15 and HERC5 were identified as interactors of the KSHV viral homolog of interferon regulatory factor 1 (vIRF1), which is known to decrease total protein ISGylation. Later studies suggested that adequate ISG15 expression can result in viral latency, while the knockdown of ISG15 and ISG20 leads to lytic reactivation of the virus [100]. A more recent study implicated CRM1 in the inhibition of KSHV by promoting nuclear retention of the autophagy adaptor protein p62 (SQSTM1) to elevate the expression of antiviral genes [99]. Interestingly, it was shown that HERC5 inhibits nuclear export of HIV RNA in a CRM1-dependent manner [59]. Although it has not been directly studied, HERC5 may inhibit the lytic phase of KSHV replication through a mechanism involving the CRM1-dependent nuclear export pathway.

Most of the viral pathogens we have discussed in this review can also resist HERC5 and ISG15 antiviral activity by employing viral protein machinery to block the host immune response. As viral–host interactions continue to progress, it is likely that more pathogens will evolve and acquire new strategies to antagonize HERC5 antiviral function. These predictions make it critical that additional studies be done to determine the mechanisms and structures that dictate the pathogen-dependent processes of HERC5 antagonism. Such knowledge will be fundamental to the development of new antiviral treatments aimed at counteracting viral antagonistic strategies employed against HERC5 in the cell.

### 4.5. Viral Antagonism of HERC5 and ISGylation

HERC5 and ISGylation are essential regulators of the antiviral immune response. Accordingly, viruses have evolved diverse strategies to disrupt HERC5 and ISG15 immune activity. For example, viral ovarian tumor domains are common among the Nairovirus family, each retaining at least a small level of deubiquitylation activity. Moreover, a pathogenic clade containing the Crimean–Congo hemorrhagic fever virus (CCHFV), Nairobi sheep disease virus (NSDV), Ganjam virus (GANV) and Erve virus (ERVEV) also possess significant deISGylation activity [126,127,128]. Experimental evidence suggests that viral ovarian tumor family-like domains (vOTUs) have adapted to deISGylate in their most common host species and that deISGylation is likely a major hurdle that viruses must cross to change host specificity [127,129]. Interestingly, the most fatal virus in humans, CCHFV, is capable of both deubiquitylating and deISGylating target proteins, likely resulting in subversion of both the antiviral and inflammatory responses [126,130]. The inhibition of the CCHFV vOTU resulted in impaired viral replication and infectivity [131,132], suggesting that this deISGylase may be a prime target for further therapeutics to treat CCHFV.

In general, it has been difficult for researchers to delineate vOTU deISGylation activity from deubiquitylation activity and to specifically measure the contribution of each process to viral fitness. However, recent structural analysis and mutagenesis studies on these viral proteins have led to the identification of the binding pocket for ISG15 and ubiquitin. Interestingly, a single amino acid substitution can change the specificity of Hazara virus from ubiquitin to ISG15, resulting in a virus that was able to selectively deISGylate while not being able to cleave ubiquitin from certain proteins [133]. These discoveries provide a useful tool for examining the effect of deISGylation and deubiquitylation in the context of innate immune activation and sensing.

Unlike the vOTUs that promiscuously bind to both ubiquitylated and ISGylated substrates, the papain-like protease of foot and mouth disease virus (FMDV), leader protease (Lb^pro^), has been shown to have stronger affinity for ISGylated proteins than for ubiquitylated proteins [91,134]. FMDV acts to cleave ISG15 molecules from the target substrate adjacent to the *C*-terminal Gly–Gly motif, which renders it unable to be recycled and attached to other target proteins [91,134]. Likewise, the loss of Lb^pro^ deISGylation activity significantly inhibited the ability of FMDV to grow in mice, although the researchers did not observe an increase in antiviral gene expression to suggest that the more likely mechanism of inhibition is through the direct ISGylation of viral proteins [91].

The inhibition of HERC5 ISGylation by influenza virus is generally species-specific. For instance, human and non-human primate ISGylation is antagonized by influenza B virus, but not in mice [124]. More recent studies have determined that influenza virus NS1B protein counteracts ISGylation by sequestering ISGylated proteins and inhibiting their incorporation in the viral ribonucleic complex [135]. Further studies are required to clarify the specific mechanisms employed by influenza viruses in different host organisms.

Intriguingly, the E3 ligase protein of vaccinia virus is a known antagonist of ISGylation [106]. When vaccinia E3 protein is expressed, there is a significant decrease in ISGylated proteins within the cell. This marked decrease corresponds with a reduction in ISGylated mitochondrial proteins and impaired mitophagy activity in the infection model. This suggests that the ISGylation of mitochondrial proteins is important for the regulation of oxidative phosphorylation and minimizing reactive oxygen species production [105].

Due to the COVID-19 pandemic, the coronavirus papain-like proteases (PL^pro^) have received considerable attention from the scientific community. Zhang et al. recently showed that expression of ubiquitin variants (UbVs) led to tight and specific binding of the PL^pro^ of Middle Eastern respiratory syndrome coronavirus (MERS-CoV) [98]. Following PL^pro^–UbV complex formation, PL^pro^ deubiquitylation, deISGylation and protease activities were significantly hindered and viral progeny were much less infectious [98]. Small-molecule inhibitor screens have subsequently identified the substrate-binding pocket and the ISG15 binding site of PL^pro^ as important determinants of viral fitness and could serve as attractive targets for antiviral drug development [136,137,138,139,140,141,142]. For example, the small molecule GRL0617 is one of the most promising SARS-CoV-2 inhibitors currently being studied that sterically blocks the binding of ISG15 and ubiquitin to PL^pro^ [137,142]. Furthermore, SARS-CoV-2 PL^pro^ has a higher affinity for ISG15 than monomeric ubiquitin and its activity is important for viral pathogenesis [143]. In cell culture and mouse models it was observed that ISGylation of the caspase-recruiting domain (CARD) of the MDA5 domain by HERC5 was important for its oligomerization and activation [104]. Knockdown of either ISG15 or MDA5 resulted in loss of IRF3 phosphorylation, which is a downstream target of MDA5 [104]. Fascinatingly, researchers have noted that naturally emerging mutations causing residue substitutions in the ISG15-binding pocket of PL^pro^ results in less pathogenic strains of SARS-CoV-2 [144]. Thus far, the primary role of PL^pro^ deISGylation appears to be suppression of the immune response, with more studies needed to determine if SARS-CoV-2 proteins are directly ISGylated by host cell machinery and if this has an overall inhibitory effect on the virus.

## 5. HERC5 and ISGylation: Moving Forward

Although it is clear that HERC5 and ISGylation represent a formidable first line of defense against viral pathogens, our knowledge of how HERC5 uses its highly conserved domain architecture to recognize and bind a diverse range of evolving viral substrates remains unclear. Future structure–function studies involving HERC5 and its protein targets will be important for determining the structural requirements necessary for these interactions. Such knowledge will help to shed light on how viral proteins evolve to evade these interactions with the host’s ISGylation machinery. Moreover, the HECT-dependent and -independent antiviral mechanisms of HERC5 raise interesting questions about the evolutionary origins of HERC5 and its biological role in vertebrate evolution. Future research in these areas will help foster the development of new therapeutic approaches using HERC5 as a prominent modulator of cellular antiviral activity to combat a wide range of infectious diseases in vertebrates.

## Figures and Tables

**Figure 1 viruses-13-01102-f001:**
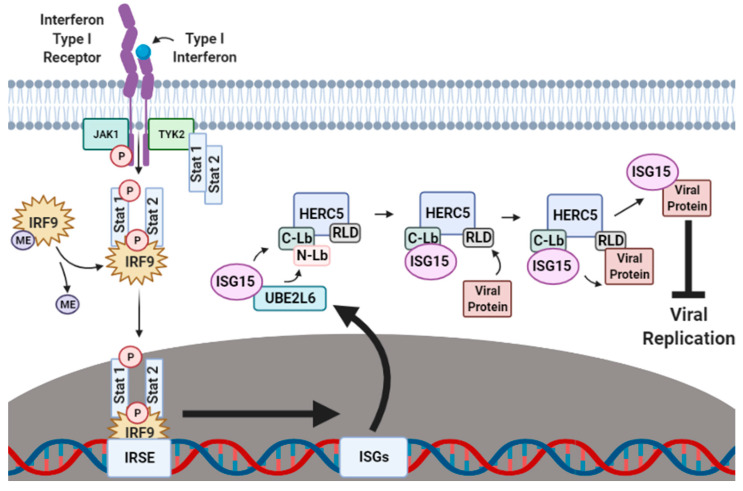
Interferon-induced ISGylation of viral proteins. During the initial stages of infection, cellular IFN-α/β surface receptors are bound by IFN-α/β-specific ligands that expose receptor phosphorylation sites to the cytoplasm. Janus kinase 1 (JAK1) and tyrosine-protein kinase 2 (TYK2) phosphorylate exposed cytoplasmic IFN-α/β receptor sites, thus recruiting the nuclear transcriptional regulators signal transducer and activator of transcription proteins 1 and 2 (STAT1 and STAT2) to also be phosphorylated by TYK2. In the cytoplasm, phosphorylated STAT1 and STAT2 form a ternary complex with methylated IRF9 which is then demethylated to signal for the migration of the STAT1–STAT2–IRF9 (SSI) complex to the nucleus. Upon nuclear entry, the SSI complex binds to the ISG promoter region interferon-stimulated response element (ISRE) to upregulate the transcription of several hundred ISGs, including ISG15, UBE1L, UBE2L6 and HERC5. Translated UBE1L, UBE2L6 and HERC5 proteins are subsequently targeted to the perinuclear regions of the cytoplasm where they work in tandem to charge, transfer and attach ISG15 onto respective viral protein substrates to inhibit viral stability, transport and reassembly. ME, methylation. This figure was created with Biorender™.

**Figure 2 viruses-13-01102-f002:**
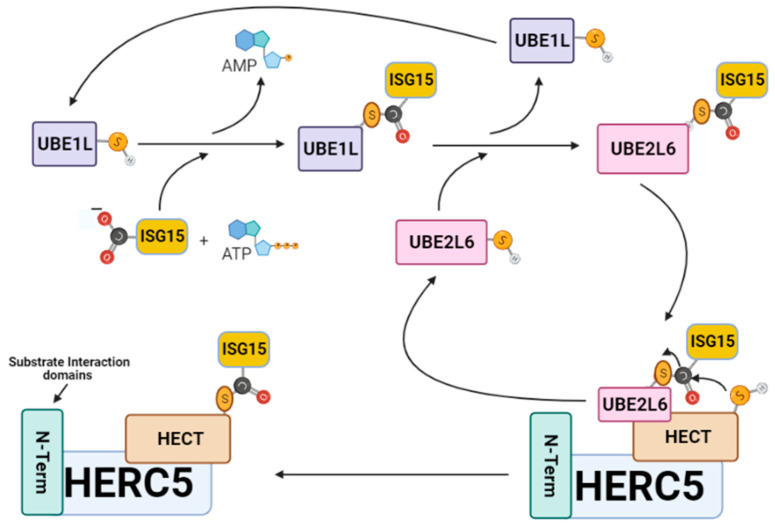
The ISGylation signaling pathway. ISG15 is sequentially transferred from UBE1L to UBE2L6, then to HERC5 before its attachment to a viral protein. HERC5 is responsible for catalyzing the specific covalent attachment of ISG15 onto viral target proteins through the formation of a stable isopeptide bond between the *C*-terminus of ISG15 and the ε-amino group on the lysine viral protein target. This figure was created with Biorender™.

**Table 1 viruses-13-01102-t001:** HERC5-dependent ISGylation of viral proteins and their downstream consequences.

Virus	Viral Protein	Viral Inhibition	Viral Antagonism	References
**Crimean–Congo hemorrhagic fever virus (CCHFV)**	Polymerase L protein	ISGylation leads to the induction of K48-dependent protein degradation	Deubiquitylase of the ovarian tumor family (OTU) removes ISG15 from proteins involved in innate immune signaling	[87,88,89]
**Ebola virus-like particles (VLP)**	Matrix protein VP40 (VP40) Viral RNA	ISG15 inhibits budding of VP40 by preventing its ubiquitylation by NEDD4 Downregulates viral RNA Inhibits viral replication	Ebola virus glycoprotein blocks HERC5 (mechanism unknown)	[56]
**Erve virus (** **ERVEV)**	Deubiquitylating ovarian tumor family (vOTU) domain protease	Effects unknown	DeISGylation of host anti-virulence factors	[90]
**Foot and mouth disease virus (FMDV)**	Leader protease (Lb^pro^)	Hypothesized to direct ISGylation of non-structural FMDV proteins	Cleaves ISG15 from ISGylated proteins prior to the C-terminus GG residues, disabling its recycling	[91,92]
**Human immunodeficiency virus-1 (HIV-1)**	Gag Viral RNA	ISGylation by HERC5 inhibits HIV viral particle production at the plasma membrane ISG15 inhibits ubiquitylation of Gag and Gag/TSG101 interaction	Effects unknown	[86]
**Human cytomegalovirus (CMV)**	Capsid scaffolding protein UL26 Intermediate–early protein 1 (IE1) Nuclear egress protein 2 (NEC2) Capsid vertex component 2 (CVC2)	ISGylation of UL26 inhibits its suppresion of NFkB ISGylation of NEC2	UL26 inhibits ISGylation IE1 reduces ISG15 transcription NEC2 downregulates UBE1L activity CVC2 prevents the degradation of UL26	[93,94,95]
**Influenza A (IAV)**	Non-structured protein 1 (NS1A)	ISGylation by HERC5 inhibits NS1 nuclear import		[96]
**Influenza B (IVB)**	Non-structured protein 1 (NS1B) Nucleoprotein (NP) Hemaglutenin protein(HA)	ISGylation inhibits the formation of infectious particles ISGylation inhibits HA protein trafficking to the cell surface	NS1B directly binds to ISG15 to inhibit HERC5 ISGylation to viral proteins NS1B binds to and sequesters ISG15-tagged proteins, limiting the incorporation of ISGylated NP protein into viral particles	[37,97,98]
**Karposi’s sarcoma herpesvirus (KSHV)**	vIRF1	ISGylation of vIRF1 by HERC5 reduces viral particle production	vIRF1 reduces ISGylation (mechanism unknown)	[99,100,101]
**Middle East respiratory syndrome coronavirus (MERS-CoV)**	Papain-like protease (PL^pro^)	Unknown	Deubiquitylase and deISGylase activity	[97,98]
**Severe acute respiratory syndrome coronavirus (SARS-CoV)**	Papain-like protease (PL^pro^)	Unknown	Deubiquitylase and deISGylase activity	[102,103]
**Severe acute respiratory syndrome coronavirus 2 (SARS-CoV-2)**	Papain-like protease (PL^pro^)	Modulation of the antiviral immune response triggered by MDA5	Cleaves ISG15 from MDA5 and other substrates	[104]
**Vaccinia virus (VACV)**	Protein E3 (p25)	ISGylation of cellular antiviral proteins	VACV protein E3 inhibits ISGylation (mechanism unknown)	[105,106,107]

## Data Availability

All discussed literature and figures are found in the main text of this mini-review.

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
