# Peer review of "HERC5 and the ISGylation Pathway: Critical Modulators of the Antiviral Immune Response"

_viruses, 2021, doi:10.3390/v13061102_

Round 1

Reviewer 1 Report

Summary

In the review “HERC5 and the ISGylation Pathway: Critical Modulators of the Antiviral Immune Response” Mathieu et al. discuss the importance of the ISGylation pathway during viral infections. The main discussion points addressed include background on the innate antiviral pathways, the ISGylation cascade, and the role of the HERC5 (an ISG15 E3 ligase) and the ISGylation pathway in regulating the host’s response to viral infection and interaction with viral proteins. Overall, the review was well written, and the figures were well designed and complement the text. The main criticism of the review includes minor disorganization of the content, particularly in the introduction and conclusion sections.

 Broad Comments

- The introduction would benefit from reordering the contents a bit so that it flows in a more logical progression through the interferon signaling pathway. The authors jump around in the pathway leaving the reader to work more to follow their meaning

- The review would benefit from adding a brief section on viral antagonism of the ISGylation pathway. At the moment it is integrated into the table, sub-section 4.4, and mentioned in the conclusion.

- The conclusion should be better developed and more focused on posing questions and how answering those questions will be beneficial. At the moment, half of the content included should be moved to the section on virus-ISGylation pathway interactions

- The figures and table are overall well-made and work well with the text.

Specific Comments

Introduction

- Unclear why using dendritic cells specifically as the example for ho the interferon induction pathway; phrased in a way that leads a less informed reader to think that this is response that occurs specifically in dendritic cells. Nothing is wrong with their explanation of the pathway though.

- Lines 45-48. Run-on sentence, consider breaking into two or rephrasing. The link between p53-dependent responses to DNA double-stranded breaks and the fact that HERC5 and ISG15 are ISGs is also not obvious and the p53-dependent aspect is not further discussed in the paragraph. I would suggest either discussing more later in the paragraph or eliminating this phrase.

- Line 56. Should be IRF9?

- Lines 54-58. This is not explained clearly. I think the author is referring to the importance of the ISGF3 complex binding to the ISRE in promoters of ISGs, but the way it is phrased now is describing IRF9 as acting on its own. Later (line 82) they refer to the ISGF3 complex as the SSI complex.

- Lines 58-60. This sentence is also phrased strangely. I think they are trying to say activation of ISGs with an ISRE in the promoter, which requires IRF9, occurs downstream of type I interferon signaling. The authors also reference a review here, not primary literature, so I’m suspicious that the author is attempting to paraphrase

- Figure 1 figure/figure legend. What does ME mean?

 - The author states twice, lines 55 and 84-85 that the ISG15 and ISGylation machinery proteins are ISGs twice. Re-ordering the phrasing would make the introduction less redundant.

The ISGylation Pathway

Line 102-105. The ISGylation pathway will also be regulating host proteins to regulate the response to viruses not just viral proteins. The authors mention that there are cellular substrates at lines 120-121.

ISG15: A Critical Moderator of the Host Antiviral Response

- Line 214. MDA is a RNA helicase

HERC5: A unique HECT E3 ubiquitin ligase that ISGylates viral proteins

- In the table, the authors include CCHFV, coronaviruses, and other viruses that antagonize ISGylation. It would be beneficial to make an additional section to discuss the antagonism of the ISGylation pathway by viral proteins separate from the antiviral functions of ISGylation.

HERC5 and ISGylation: Moving forward

Lines 408-417. Unclear why the authors are discussing the specific examples that could have been included in the proceeding section in the conclusion paragraph. I think it would be beneficial to discuss the open questions remaining regarding ISGylation in the antiviral response during the conclusion.

It would be beneficial to spell out/list specifically the questions that are most pressing for research to address. The author refers to “questions” (Lines 417 and 425), but doesn’t actually state clearly what their questions are.

Author Response

Response to Reviewer 1's comments:

Summary

In the review “HERC5 and the ISGylation Pathway: Critical Modulators of the Antiviral Immune Response” Mathieu et al. discuss the importance of the ISGylation pathway during viral infections. The main discussion points addressed include background on the innate antiviral pathways, the ISGylation cascade, and the role of the HERC5 (an ISG15 E3 ligase) and the ISGylation pathway in regulating the host’s response to viral infection and interaction with viral proteins. Overall, the review was well written, and the figures were well designed and complement the text.

We thank the reviewer for these compliments.

The main criticism of the review includes minor disorganization of the content, particularly in the introduction and conclusion sections.

We thank the reviewer for this constructive feedback.  We have gone ahead and made all of the suggested revisions (please see comments below).

 Broad Comments

- The introduction would benefit from reordering the contents a bit so that it flows in a more logical progression through the interferon signaling pathway. The authors jump around in the pathway leaving the reader to work more to follow their meaning

We thank the reviewer for this suggestion.  We have re-organized our introduction to improve clarity.

- The review would benefit from adding a brief section on viral antagonism of the ISGylation pathway. At the moment it is integrated into the table, sub-section 4.4, and mentioned in the conclusion.

We thank the review for this helpful suggestion.  We agree and have added a new section to our review on “Viral Antagonism of HERC5 and ISGylation.”

The conclusion should be better developed and more focused on posing questions and how answering those questions will be beneficial. At the moment, half of the content included should be moved to the section on virus-ISGylation pathway interactions

We thank the reviewer for this constructive feedback.  We have modified our conclusion.

- The figures and table are overall well-made and work well with the text.

We thank the reviewer for this compliment.

Specific Comments

Introduction

- Unclear why using dendritic cells specifically as the example for ho the interferon induction pathway; phrased in a way that leads a less informed reader to think that this is response that occurs specifically in dendritic cells. Nothing is wrong with their explanation of the pathway though.

We agree with the reviewer. We have changed “dendritic cells” to “immune cells”.

 - Lines 45-48. Run-on sentence, consider breaking into two or rephrasing. The link between p53-dependent responses to DNA double-stranded breaks and the fact that HERC5 and ISG15 are ISGs is also not obvious and the p53-dependent aspect is not further discussed in the paragraph. I would suggest either discussing more later in the paragraph or eliminating this phrase.

We thank the reviewer for this suggestion. We agree and have removed the p53 reference.

- Line 56. Should be IRF9?

We thank the reviewer for pointing this out.  We have corrected the statement.

- Lines 54-58. This is not explained clearly. I think the author is referring to the importance of the ISGF3 complex binding to the ISRE in promoters of ISGs, but the way it is phrased now is describing IRF9 as acting on its own. Later (line 82) they refer to the ISGF3 complex as the SSI complex.

Thank you for pointing this out.  We have clarified these statements in the text.

- Lines 58-60. This sentence is also phrased strangely. I think they are trying to say activation of ISGs with an ISRE in the promoter, which requires IRF9, occurs downstream of type I interferon signaling. The authors also reference a review here, not primary literature, so I’m suspicious that the author is attempting to paraphrase

Thank you. We have now clarified this in the text.

- Figure 1 figure/figure legend. What does ME mean?

We thank the reviewer for pointing this out.  We have now defined “ME” in the legend as methylation.

 - The author states twice, lines 55 and 84-85 that the ISG15 and ISGylation machinery proteins are ISGs twice. Re-ordering the phrasing would make the introduction less redundant.

Thanks for the suggestion. We have re-ordered the phrasing.

The ISGylation Pathway

Line 102-105. The ISGylation pathway will also be regulating host proteins to regulate the response to viruses not just viral proteins. The authors mention that there are cellular substrates at lines 120-121.

We have made the correction.

ISG15: A Critical Moderator of the Host Antiviral Response

- Line 214. MDA is a RNA helicase

Corrected. Thank you.

HERC5: A unique HECT E3 ubiquitin ligase that ISGylates viral proteins

- In the table, the authors include CCHFV, coronaviruses, and other viruses that antagonize ISGylation. It would be beneficial to make an additional section to discuss the antagonism of the ISGylation pathway by viral proteins separate from the antiviral functions of ISGylation.

We agree with the reviewer.  We have now added a section “Viral Antagonism of HERC5 and ISGylation” where we discuss how other viruses antagonize ISGylation.

HERC5 and ISGylation: Moving forward

Lines 408-417. Unclear why the authors are discussing the specific examples that could have been included in the proceeding section in the conclusion paragraph. I think it would be beneficial to discuss the open questions remaining regarding ISGylation in the antiviral response during the conclusion.

It would be beneficial to spell out/list specifically the questions that are most pressing for research to address. The author refers to “questions” (Lines 417 and 425), but doesn’t actually state clearly what their questions are.

We thank the reviewer for these helpful suggestions. We have removed the examples in the first part of the conclusion and identified more specific questions for future thought.

Reviewer 2 Report

In this review, the authors introduced the general pathway of ISGylation and its regulation. They then focused on the key E3 enzyme, HERC5, to discuss the anti-viral effects of ISGylation as well as several examples of viral counteractions. The outline of manuscript is logical and clear. A few minor improvements are needed before publication.

  • Much of the sections are dedicated to introducing the HERC5/ISGylation regulation in HCV, HIV and IAV, which are well characterized in the past 10-15 years. However, the newer findings in CMV and vaccinia virus were only mentioned with little details. Some of the viruses listed in the table were not mentioned at all. It would be helpful if the recent discoveries are explained in more details.
  • The list of viruses in Table 1 seems random. It would be easier to follow if the authors make it in alphabetical order of the virus or in chronological order of the discoveries, or categorize the viruses according to family. More importantly, coordinate the information summarized in Table 1 with the text. For example, HCV is not found in the table.
  • In section 4.3, the short description of ISGylation in HIV at the beginning of the paragraph seems repetitive and confusing, since much of the details are introduced right after.
  • In lines 282-284, HERC6 is mentioned in one sentence. The information introduced by this sentence is not clear.
  • Lines 410-411: I don’t understand this sentence about ebolavirus VP40. Lines 182-184: I don’t understand this one either.
  • Line 56: IRF9 instead of IF9? Line 321: IAV instead of HCV? Line 421: EVD is the disease, not name for the virus.

Author Response

In this review, the authors introduced the general pathway of ISGylation and its regulation. They then focused on the key E3 enzyme, HERC5, to discuss the anti-viral effects of ISGylation as well as several examples of viral counteractions. The outline of manuscript is logical and clear.

We thank the reviewer for these compliments.

A few minor improvements are needed before publication.

  • Much of the sections are dedicated to introducing the HERC5/ISGylation regulation in HCV, HIV and IAV, which are well characterized in the past 10-15 years. However, the newer findings in CMV and vaccinia virus were only mentioned with little details. Some of the viruses listed in the table were not mentioned at all. It would be helpful if the recent discoveries are explained in more details.

We thank the reviewer for this helpful suggestion.  We agree and have expanded on this with a new section in our review on ““Viral Antagonism of HERC5 and ISGylation.”

  • The list of viruses in Table 1 seems random. It would be easier to follow if the authors make it in alphabetical order of the virus or in chronological order of the discoveries, or categorize the viruses according to family. More importantly, coordinate the information summarized in Table 1 with the text. For example, HCV is not found in the table.

We thank the reviewer for this constructive feedback.  As suggested, we have listed the viruses in alphabetical order and we now discuss each virus listed in the table within the text of our review.

  • In section 4.3, the short description of ISGylation in HIV at the beginning of the paragraph seems repetitive and confusing, since much of the details are introduced right after.

We thank the reviewer for pointing this out.  We have modified the text.

  • In lines 282-284, HERC6 is mentioned in one sentence. The information introduced by this sentence is not clear.

Thank you for pointing this out.  We have modified the text.

  • Lines 410-411: I don’t understand this sentence about ebolavirus VP40.

We have expanded and clarified this sentence.

  • Lines 182-184: I don’t understand this one either.

We thank the reviewer for this constructive feedback.  We have removed this sentence.

  • Line 56: IRF9 instead of IF9?

We have corrected this typo.

  •  Line 321: IAV instead of HCV?

We thank the reviewer for pointing this out.  We have corrected this statement.

  • Line 421: EVD is the disease, not name for the virus.

We have corrected this sentence.